# Steam Reforming of Chloroform-Ethyl Acetate Mixture to Syngas over Ni-Cu Based Catalysts

Qiong Wu, Chenghua Xu *, Yuhao Zheng, Jie Liu *, Zhiyong Deng and Jianying Liu

College of Resources and Environment, Chengdu University of Information and Technology, Chengdu 610225, China; WuQiongself@163.com (Q.W.); zhengyuhao0220@163.com (Y.Z.); dengzhiyong@cuit.edu.cn (Z.D.); liujy@cuit.edu.cn (J.L.)

* Correspondence: xch@cuit.edu.cn (C.X.); ljzyhj@cuit.edu.cn (J.L.)

**Abstract:** NiCuMoLaAl mixed oxide catalysts are prepared and applied in the steam reforming of chloroform-ethyl acetate (CHCl$_3$-EA) mixture to syngas in the present work. The pre-introduction of Cl- ions using chloride salts as modifiers aims to improve the chlorine poisoning resistance. Catalytic tests show that KCl modification is obviously advantageous to increase the catalytic life. The destruction of catalyst structure induced by in situ produced HCl and carbon deposits that occurred on acidic sites are two key points for deactivation of reforming catalysts. The presence of Cl$^-$ ions gives rise to the formation of an Ni-Cu alloy, which exhibits a synergetic effect on catalyzing reforming along with metallic Ni crystals formed from excess nickel species, and giving an excellent catalytic stability. Less CHCl$_3$ and more steam can also increase the catalytic stable time of KCl-modified NiCuMoLaAl reforming catalyst.

**Keywords:** resource utilization of organic waste liquid; steam reforming; resistance to chlorine poisoning; chloride salts modification





## 1. Introduction

With the rapid development of industrialization and urbanization, there are large amounts of organic waste liquid discharged from organic synthesis labs and many productive processes involved in the petrochemical, papermaking, leather, food, and pharmaceutical industries [1–3]. The discharged organic waste liquid has a wide range of sources, complex components, high chromaticity, and heavy odor [4], which can not only damage the environment, but eventually endanger humans and earth creatures through bioaccumulation.

At present, some available methods, including physical, biological, and chemical oxidation, have been applied in treating organic waste. For example, some physical technologies such as adsorption, precipitation, filtration, and ultrasound are used to remove organic compounds in wastewater [5,6]. Biological and chemical oxidation, such as the Fenton-like process, incineration, and electrochemical and photocatalytic oxidation [7–13], are applied to degrade organic substances dissolved in wastewater. As is well known, organic waste liquid contains rich carbon and hydrogen resources. However, the above treatment technologies have no obvious advantages on resource utilization of organic waste. Driven by energy and environmental issues, catalytic steam reforming (CSR) has been widely used in recycling organic materials to produce clean energy hydrogen and high-value syngas because of its high effectiveness and economic feasibility [14–16].

Noble metal (Pt, Ru, and Pd) supported catalysts have been found to exhibit excellent activity and stability for CSR of alcohols, alkanes, and their mixed system [17–19]. Considering catalyst cost, many non-noble metals such as Ni, Cu, and Co-based catalysts have been explored and applied in the production of hydrogen and syngas through CSR [20,21]. Of course, the catalysts simultaneously containing noble and transition metals as an active phase are also used in steam reforming [22,23]. Among these catalysts, Ni-based catalysts

are widely investigated due to their excellent activity and relatively low cost [24–27]. However, active catalytic Ni species are found to succumb to sintering at a high temperature, leading to deactivation during reforming [28], which can be inhibited by the incorporation of Cu [21,29]. In order to obtain Ni-based reforming catalysts with a good stability, Mo and rare earth La oxide as a promoter have been introduced. They can disperse active metal species of the catalyst and provide medium and strong basic sites, which are beneficial to increasing catalytic activity and inhibiting coking during the steam reforming of organic substances [16,30–32]. As is well known, the composition of organic waste is complex and often contains N, S, and Cl-heteroatom chemicals, aromatic compounds, and toxic organic substances [33]. In particular, the presence of heteroatom-containing chemicals will probably lead to catalyst rapid deactivation during CSR. For example, HCl as a by-product of the steam reforming of chlorocarbons contains a Cl element that easily reacts with the catalytic active phase. From the investigation on CSR of methyl chloride and chlorinated aromatics over Ni-based catalysts as earlier reported [17,34–38], it has been concluded that the deactivation of reforming catalysts mainly results from the following issues: (1) the produced HCl can react with catalysts support metal oxides; (2) the presence of HCl leads to the sintering of catalytic active phase with support; (3) HCl adsorbed on catalyst surface can provide acidic sites resulting in carbon deposits. Therefore, it is important for the steam reforming of chlorocarbon-containing organic waste liquid to improve its ability to resist Cl-poisoning of catalysts. Although noble metals and high-loading Ni catalysts exhibit a better resistance to HCl poisoning, the destruction of the catalyst structure cannot be avoided due to HCl reaction with support. Other previous studies [39,40] have revealed that pre-treatment using HCl before catalytic reaction can prevent catalysts from reacting to HCl generated during hydrodechlorination of chlorobenzene and other complex chlorocarbons.

Therefore, the present work prepares a NiCuMoLaAl complex oxide precursor by using co-precipitation according to the catalyst system used in glycerol steam reforming reported in our previous work [41], and followed by pre-introducing chloride salts to obtain $Cl^-$ ion-incorporated reforming catalysts through impregnation. The presence of $Cl^-$ ions in the catalyst is expected to inhibit the rapid deactivation from adsorption of and reaction with HCl produced during reforming. The effect of introduced chloride salts on physicochemical and catalytic performance in steam reforming of $CHCl_3$-EA mixture is also investigated. It aims to explore the function of $Cl^-$ ions in catalytic active phase formation, acid-base property, and resistance to HCl-poisoning of catalyst during steam reforming of $CHCl_3$ and EA mixture ($CHCl_3 + H_2O \rightarrow CO + 3HCl$ and $C_4H_8O_2 + 2H_2O \rightarrow 4CO + 6H_2$), and obtain a potential catalyst to apply to resource utilization of chlorocarbon-containing organic waste liquid to $H_2$ or syngas through steam catalytic reforming.

## 2. Results and Discussion

### 2.1. Catalytic Performance

From Table 1, it can be clearly observed that all prepared NiCuMoLaAl mixed oxide catalysts (NCML) exhibit high catalytic activity in steam reforming of $CHCl_3$-EA mixture, almost all organic reactants are converted to gaseous products. Moreover, the collected liquid products over all catalysts are observed to be colorless and clear liquid. Steam reforming gives a conversion of about 99.46% organic reactant and has a gas production capacity of 4.07 $m^3$ $kg_{oil}^{-1}$ over NCML without any modifier. For the catalysts modified by chloride salts, such as $NH_4Cl$, NaCl, or KCl, the conversion of organic reactant is decreased to some extent. However, the gas production capacity of NCML catalyst is increased by modification of chloride salts. From gaseous product distribution, the reforming gas obtained over all catalysts mainly contains about 70% $H_2$, 12–16% CO, 14–16% $CO_2$, and 1–4% $CH_4$. From Table 1, NCML catalysts modified by chloride salts are found to exhibit a higher $H_2$ and $CO_2$ selectivity, and a lower CO selectivity than NCML without any modifiers in steam reforming of $CHCl_3$-EA. It is mainly because during the CSR of $CHCl_3$-EA, the formed CO can cause a water-gas shift reaction to generate $CO_2$, which will give

rise to the formation of more $H_2$. Therefore, the modification with chloride salts is proven to improve the catalytic ability of the catalyst for water-gas shift reactions (CO + $H_2O$ ↔ $CO_2$ + $H_2$) resulting in a higher gas production capacity.

**Table 1.** Catalytic performance of modified catalysts in steam reforming of organic waste liquid *.

| Catalysts | Conversion of Oil (%) | Capacity of Gas Production ($m^3 \cdot kg_{oil}^{-1}$) | Catalytic Life (min) | Carbon Deposit ($mg\ g_{cat}^{-1} \cdot h^{-1}$) | Gaseous Products Distribution (mol%) | | | |
|---|---|---|---|---|---|---|---|---|
| | | | | | $H_2$ | CO | $CO_2$ | $CH_4$ |
| NCML | 99.46 | 4.07 | 248 | 43.7 | 68.25 | 16.36 | 14.27 | 1.12 |
| $NH_4Cl$/NCML | 98.83 | 4.52 | 272 | 49.6 | 70.07 | 13.16 | 15.16 | 1.61 |
| NaCl/NCML | 98.30 | 4.55 | 291 | 42.4 | 69.27 | 12.15 | 15.21 | 3.37 |
| KCl/NCML | 97.83 | 4.35 | 408 | 40.7 | 69.83 | 12.85 | 14.77 | 2.55 |
| KCl/NCML @ | 97.97 | 4.27 | 638 | 38.0 | 69.68 | 13.15 | 14.06 | 3.11 |

* Catalyst reduction at 800 °C with a heating rate of 8 °C·min$^{-1}$, reforming temperature 750 °C, liquid hourly space velocity (LHSV, pumped volume (ml) of $CHCl_3$-EA mixture liquid per hour per ml catalyst) 3.3 h$^{-1}$, $H_2O$/C molar ratio 2.13, chlorine content of $CHCl_3$-EA 67,800 ppm, 80 mL·min$^{-1}$ $N_2$ as balance gas; @ chlorine content of $CHCl_3$-EA 33,900 ppm.

From Figure 1, it can clearly be found that the capacity of gas production from CSR of $CHCl_3$-EA over all catalysts rapidly increases from about 3.6 to 4.2 $m^3 \cdot kg_{oil}^{-1}$ when CSR starts about 20 min, indicating that there is a catalytic activation process during the initial reforming stage. It is mainly due to the presence of reducing atmosphere provided by the produced $H_2$ from CSR. NaCl/NCML catalysts exhibit another increased stage in their capacity to produce gas after reforming 130 min, showing the reforming active sites are increased due to further exposure to the reducing atmosphere. Meanwhile, from Table 1 and Figure 1 it is also observed that the NCML catalyst without the chloride salt modification sees a sharp decrease in its capacity to produce gas after about 4 h of steam reforming. As is well known, $CHCl_3$ molecules in $CHCl_3$-EA mixtures contain a Cl element, which can be converted to HCl during steam reforming. The formed HCl will give rise to two possible negative effects: (1) reactions with a catalytic active phase leading to the destruction of the catalyst structure; (2) more acidic sites leading to carbon deposition on the catalyst's surface. All these can result in the catalyst deactivation during reforming. If the deactivation of catalysts results from the destruction of catalyst structure by HCl, the production of gaseous products should exhibit a gentle decrease. However, it is found from Figure 1 that CSR of $CHCl_3$-EA over all catalysts gives a rapid termination, indicating that acidic species provided by the formed HCl result in the occurrence of carbon deposit. The modification of chloride salts such as $NH_4Cl$, NaCl, and KCl can prolong the catalytic life of NCML to 272 min, 291 min, and 408 min, respectively. It shows that the introduced chloride salts can improve the catalytic stability of catalysts in the CSR of $CHCl_3$-EA. Moreover, it is also found that the KCl-modified NCML catalyst exhibits an excellent catalytic stability, and still gives a longer stable catalytic time of 638 min in the reforming of organic mixture with a lower chlorine content. It further proves that the formed HCl plays an important role in affecting the catalyst's stability.

## 2.2. Textural and Structure Properties

The $N_2$ adsorption–desorption results of the catalysts are shown in Table 2. Obviously, specific surface area, pore volume, and pore diameter of NCML reforming catalysts are increased by the addition of chloride salts. It shows that the introduction of chloride salts does not give rise to the blockage effect, but possibly results in the change of catalyst structure during calcination [42,43]. XRD results of reduced catalysts (Figure 2) show that aluminum oxide in the catalyst mainly exists in the form of γ-$Al_2O_3$ (pdf No. 1-1308) acting as a support. The fresh reduced NCML without chloride salt modification exhibits some weak metal $Ni^0$ diffraction peaks (pdf No. 1-1258) at 2θ = 44.7°, 51.7°, and 71.6°, respectively. It shows that nickel species in NCML without chlorinated salts mainly exist in a solid solution through reaction with Mo, La, and Al oxides during catalyst calcination [32,44], which $H_2$ struggles to reduce.

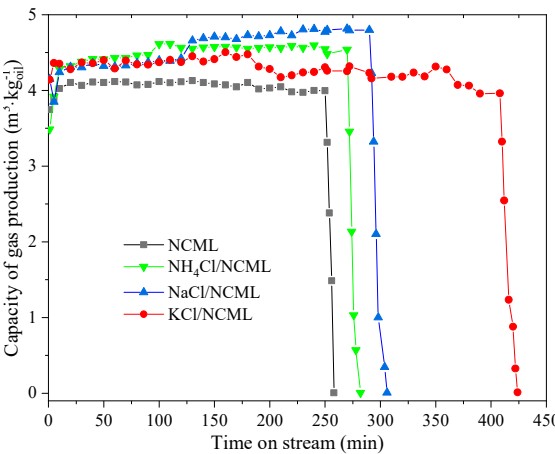

**Figure 1.** Stability of gas production from steam reforming of CHCl₃-EA mixture over different catalysts. Reforming conditions: temperature 750 °C, LHSV 3.3 h$^{-1}$, chlorine content in organic waste liquid 67800 ppm, H₂O/C molar ratio 2.13, balance gas 80 mL·min$^{-1}$ N₂.

**Table 2.** The physical properties of different catalysts.

| Catalysts | N₂ Adsorption–Desorption | | | Acidic Sites (mmol · g$_{cat}^{-1}$) [a] | | | Cell Size of Nickel [b] | | |
|---|---|---|---|---|---|---|---|---|---|
| | $S_{BET}$ (m²·g$^{-1}$) | $V_{BJH}$ (mL·g$^{-1}$) | $D_{pore}$ (nm) | $A_W$ | $A_M$ | $A_T$ | $d_{Ni,F}$ (nm) | $d_{Ni,S}$ (nm) | $\Delta d_{Ni}$ (nm) |
| NCML | 87.91 | 0.40 | 6.25 | 0.038 | 0.173 | 0.211 | 8.10 | 22.89 | 14.78 |
| NH₄Cl/NCML | 92.83 | 0.51 | 6.83 | 0.106 | 0.112 | 0.218 | 14.61 | 23.23 | 8.63 |
| NaCl/NCML | 94.37 | 0.52 | 6.65 | 0.039 | 0.090 | 0.128 | 17.08 | 22.02 | 4.94 |
| KCl/NCML | 95.04 | 0.46 | 6.34 | 0.036 | 0.071 | 0.107 | 18.44 | 24.21 | 6.78 |

[a] Calculated by NH₃-TPD results of catalyst reduced at 800 °C for 2 h. $A_W$—amount of weak acidic sites, $A_M$—amount of medium-strong acidic sites, $A_T$—amount of total acid sites per gram catalyst. [b] Calculated by Scherrer formula from XRD results of reduced samples, *F* fresh catalyst, *S* spent catalyst. $d_{Ni,F}$—Ni cell size of fresh catalyst, $d_{Ni,F}$—Ni cell size of spent catalyst, $\Delta d_{Ni}$—growth of Ni cell size.

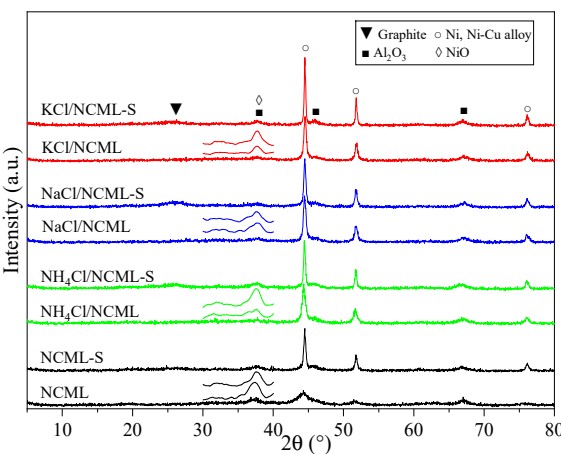

**Figure 2.** XRD patterns of fresh and spent (S) catalysts reduced at 800 °C.

From Figure 2, all NCML catalysts modified by chloride salts including NH₄Cl, NaCl, and KCl are clearly observed to exhibit obvious diffraction peaks characteristic of metallic nickel crystal (pdf No. 65-380) or Ni-Cu alloy (pdf No. 65-9047) at 2θ = 44.7°, 51.8°, and 76.1° [45]. Moreover, the intensity of the diffraction peak at 2θ = 37.7° is characteristic of NiO (pdf No. 2-1216). Furthermore, γ-Al₂O₃ is weakened by the modification of the chloride salts. It indicates that the modification of the chloride salts leads to the agglomeration of highly dispersed nickel and copper species or migration of NiO and CuO species from solid solution, resulting in the formation of bigger metallic nickel or Ni-Cu alloy particles after reduction by H₂ [46]. From Table 2, the size of nickel crystals in the reduced catalyst is also found to increase as a modifier order of none < NH₄Cl < NaCl <

KCl, which is consistent with the catalytic life order of catalysts (see Table 1). Therefore, the modification of chloride salts results in Ni crystal or Ni-Cu alloy growth of catalysts, and big metallic nickel or Ni-Cu alloy particles exhibit a catalytic stability in steam reforming of $CHCl_3$-EA.

From the XRD patterns of the spent catalysts (Figure 2), the NCML catalyst is observed to give obvious diffraction peaks of metallic Ni or Ni-Cu alloy after steam reforming. In fact, the steam reforming of $CHCl_3$-EA mixture has produced gases of $H_2$, CO, $CO_2$, and HCl, providing a reducing atmosphere. The NiCuMoLa mixed oxide catalysts are easily destroyed to form dissociated nickel and copper species such as $NiCl_2$, $CuCl_2$, NiO, or CuO, free of the catalyst phase, which is on-line reduced by the formed $H_2$ to metallic Ni crystal or Ni-Cu alloy [34]. Of course, occurrence of this process easily gives rise to the rapid deactivation of NCML without chloride salts modification in the steam reforming. On the other hand, although the spent NCML catalysts modified by $NH_4Cl$, NaCl, and KCl also give increasing diffraction peaks characteristic of metallic Ni, the growth of nickel crystalline particles is not too obvious (see Figure 2 and Table 2). Therefore, the modification of chloride salts is deduced to act as preventing the destruction of the catalyst structure, which can improve the life of catalysts in steam reforming. However, it has been found from catalytic testing that the stability of catalysts modified by chloride salts still does not reach an ideal level for the steam reforming of $CHCl_3$-EA mixture. In fact, deposited carbon, which possibly comes from direct cleavage of organic reactants ($2CHCl_3 \rightarrow 2C + H_2 + 3Cl_2$, $C_4H_8O_2 \rightarrow 2C + 2H_2O + 2H_2$) or CO disproportionation reaction ($2CO \rightarrow C + CO_2$) has also been found over the spent catalysts by the XRD result (pdf No. 3-401) peak at $2\theta = 26.5°$ [47]. These further prove that the main factor for affecting the catalytic stability is the carbon deposit that occurs on acidic sites provided by the adsorbed HCl, which can be inhibited by the pre-introduced $Cl^-$ ions to some extent.

### 2.3. Reductive Property

From $H_2$-TPR profiles (Figure 3), two obvious reduction peaks for the NCML catalyst are observed at about 270 °C and 800 °C, respectively. The low-temperature reduction peak is assigned to the reduction of highly dispersed NiO, CuO, or Ni-Cu mixed oxides to metallic Ni or Ni-Cu alloys [21]. The high-temperature reduction peak at about 800 °C is attributed to the reduction of $Ni^{2+}$ species located in the $NiAl_2O_4$ or $NiO-Al_2O_3$ solid solution [32], which exhibits a strong interaction between metal and supports (SIMS). Meanwhile, the modification of chloride salts such as $NH_4Cl$, NaCl and KCl is found to make NCML catalysts exhibit two reduction peaks in the low-temperature range of 200–400 °C. The $H_2$-consumption peak at about 269 °C should be assigned to the reduction of CuO to $Cu^0$ [48], and another one at about 340 °C possibly results from the reduction of NiO dispersed over the catalyst surface [16]. It shows that the highly dispersed nickel and copper oxide species over the NCML catalyst mainly exist in the form of NiO-CuO mixed oxides [21]. The modification of chloride salts can lead to the disassociation of mixed oxides to isolated NiO and CuO species. Moreover, three $H_2$-consumption peaks, assigned to the reduction of CuO, NiO, and $Ni^{2+}$ in $NiAl_2O_4$ or $NiO-Al_2O_3$ solid solution, shift to lower temperature as the order of $NH_4Cl$ < NaCl < KCl, which is in agreement with the catalytic life of modified NCML catalysts (see Table 1 and Figure 1). Combined with the XRD results, the reduction order of active components in NCML catalysts modified by chloride salts should be highly dispersed CuO, NiO, and $Ni^{2+}$ located in $NiAl_2O_4$ or $Ni-Al_2O_3$ solid solution during catalyst reduction by $H_2$. Therefore, it can be concluded that modification of chloride salts promotes the easy reduction of $Ni^{2+}$ and $Cu^{2+}$ to the metallic phase, which acts as a catalytic site for steam reforming.

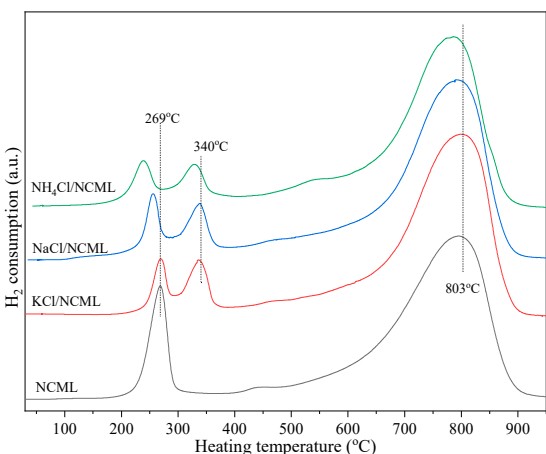

**Figure 3.** H$_2$-TPR profiles of catalysts with and without Cl$^-$ ion modification.

### 2.4. Acid-Base Property

From the NH$_3$-TPD results (Figure 4a), it can be observed that the four catalysts have two obvious NH$_3$-desorption peaks. The low-temperature peak at about 190 °C is assigned to NH$_3$ desorption over weak acidic sites, while the one at about 500 °C is from medium-strong acidic sites [49]. NH$_4$Cl modification will make both desorption peaks strong and shift to high temperature. This is due to the fact that the introduced NH$_4$Cl can be decomposed to HCl, which interacts with the oxide phase in catalysts to provide more acidic sites during calcination. Although the presence of Cl$^-$ ions is helpful to the formation of stable Ni metallic and Ni-Cu alloy active centers, the formation of more acidic sites will be disadvantageous to increase catalytic stability due to the possible carbon deposit. This is just the reason for no obvious improvement in the catalytic stability of NH$_4$Cl-modified NCML (see Table 1 and Figure 1). Moreover, the introduction of alkali metal chloride salts is found to make both NH$_3$-desorption peaks small and shift to lower temperature, indicating that acidic sites and acidity of NCML reforming catalysts are decreased. From the CO$_2$-TPD results (Figure 4b), for chloride salt-modified NCML catalysts CO$_2$-desorption peaks assigned to weak base sites shift to much lower temperatures and their intensities decrease. Although the high-temperature CO$_2$-desorption peaks also shift to lower temperature, their intensities increase. These indicate that the introduction of chloride salts leads to the formation of less weak base sites and more medium-strong base sites over the catalyst surface.

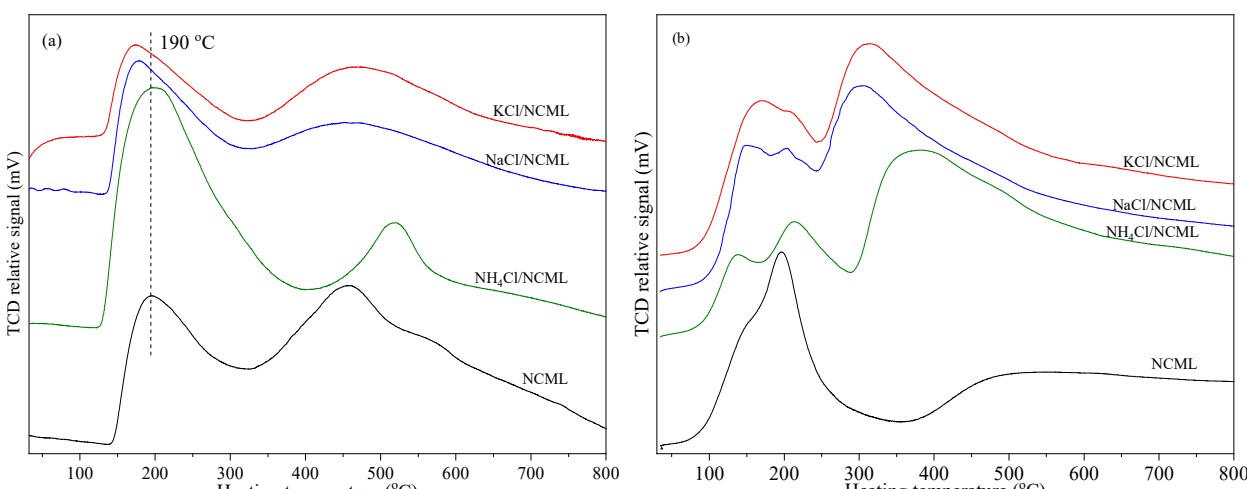

**Figure 4.** NH$_3$-TPD (**a**) and CO$_2$-TPD (**b**) curves of reduced catalysts.

### 2.5. SEM Results

The surface morphology of fresh NCML catalysts were obtained by SEM, and the results are shown in Figure 5. It is seen that the surface of fresh catalysts modified with chloride salts are distinctly different from that of NCML without any modifier. HCl from NH$_4$Cl decomposition can destroy NCML morphology (Figure 5b). Modification of NaCl and KCl (Figure 5c,d) is obviously found and results in the formation of isolated Ni crystals and Ni-Cu alloy particles over the reduced NCML catalysts.

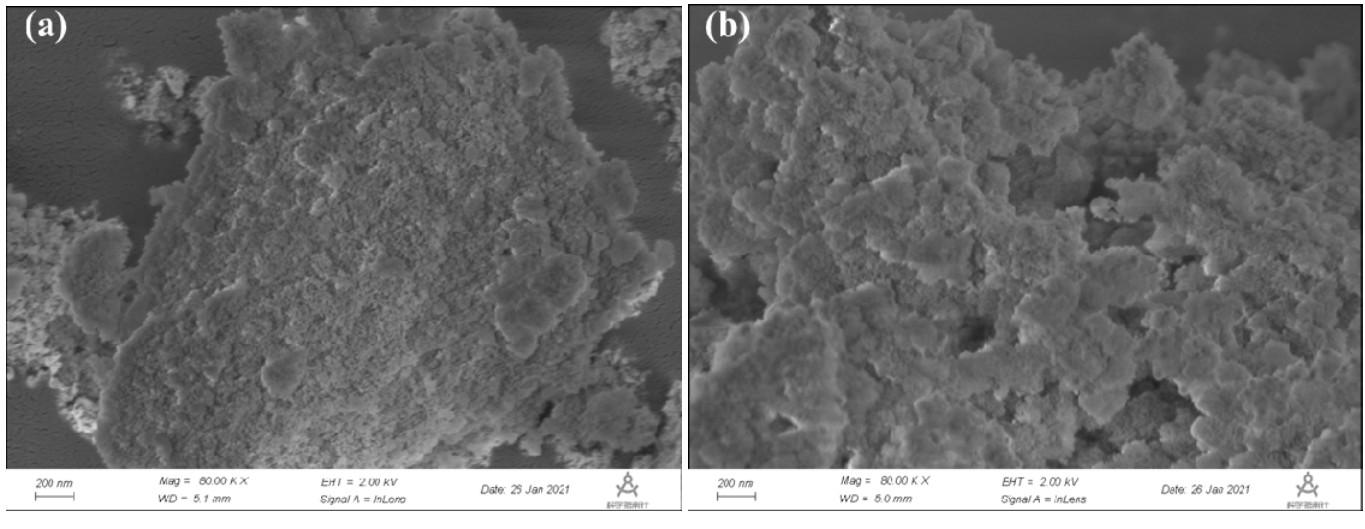

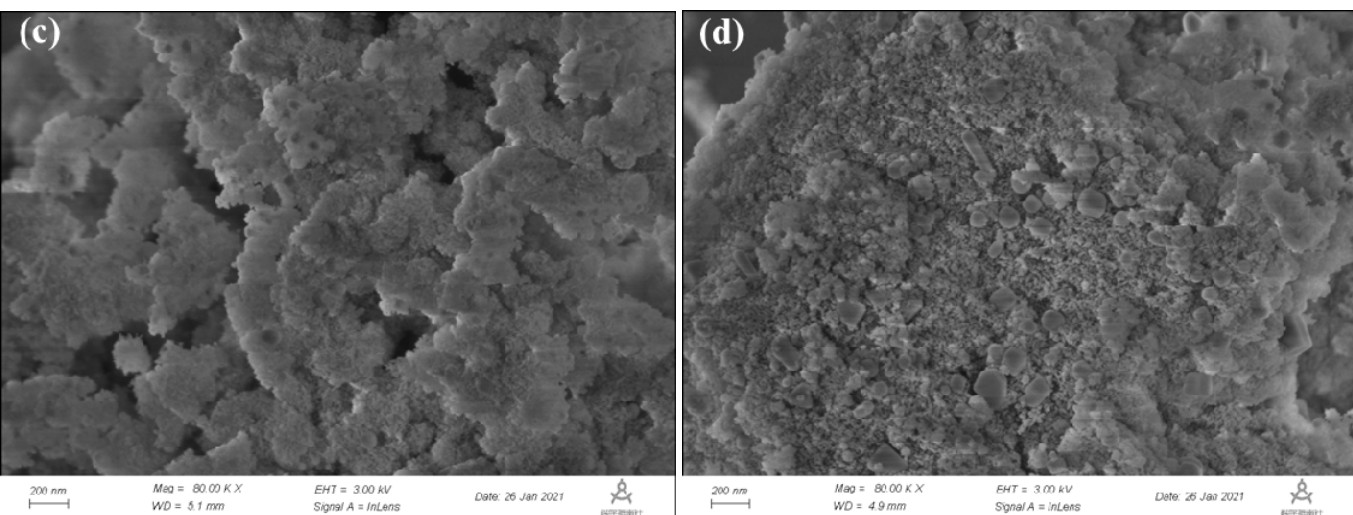

**Figure 5.** SEM images of fresh reduced NCML catalysts modified by none (**a**), NH$_4$Cl (**b**), NaCl (**c**), and KCl (**d**).

From SEM results of spent NCML catalysts (Figure 6), NCML without any modifier gives abundant slim filamentous coke covering its surface after reforming (Figure 6a). The formation of pulverized surface is observed for the spent NH$_4$Cl-modified NCML, perhaps resulting from the partial destruction of the structure. From Table 1 and Figure 1, NH$_4$Cl/NCML also exhibits a poor catalytic life and carbon deposit similar to NCML without chloride salts. Therefore, carbon deposit and structure damage are concluded to be two key factors affecting the reforming life of catalysts. Carbon species deposited on NaCl-modified NCML (Figure 6c) are discovered to exhibit a rough filamentous coke, which is due to the fact that NaCl introduction decreases the acidity of the catalyst surface. However, compact spherical carbon particles are deposited on surface of deactivated KCl-modified NCML (Figure 6d), which will produce major gas resistance, which leads to

catalyst deactivation during reforming [50]. Combined with the catalytic life of catalysts (Table 1), it can be deduced that KCl modification has no damage catalyst structure, but cannot completely avoid the carbon deposit.

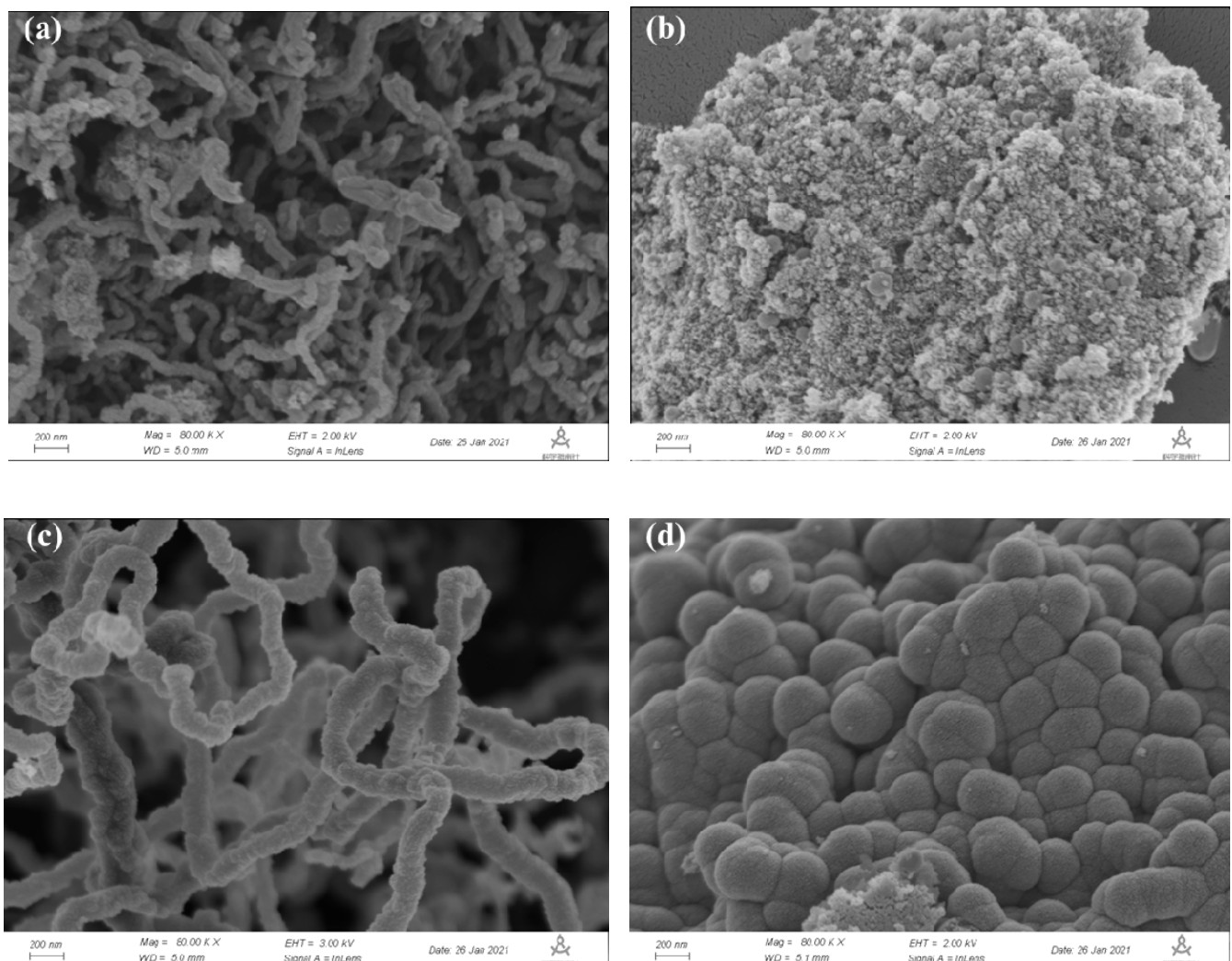

**Figure 6.** SEM images of spent NCML catalysts modified by none (**a**), NH$_4$Cl (**b**), NaCl (**c**), and KCl (**d**).

### 2.6. Effect of Reforming Parameters

According to results obtained in Section 3.3, a low heating rate during catalysts reduction easily leads to complete reduction of Cu$^{2+}$ to Cu$^0$ in the low-temperature range, resulting in the production of Cu crystal cores. From Table 3, it can be found that the heating rate during catalysts reduction has no obvious effect on the catalytic activity and gas production capacity of KCl/NCML in reforming of CHCl$_3$-EA mixture. However, a low heating rate will make KCl/NCML exhibit a longer catalytic life and produce more CH$_4$ along with a decreasing H$_2$ in reforming. It shows that Ni metallic particles and Ni-Cu alloy exhibit excellent catalytic reforming stability due to the presence of the metal–metal interface. Of course, the simultaneous formation of Ni metallic particles and Ni-Cu alloy probably provides catalytic sites for methanation.

Moreover, the increase in liquid hourly space velocity (LHSV) will give rise to a decrease in catalytic stable time and conversion of organic substance CHCl$_3$ and EA. It is mainly because the contact time between reactants and catalytic active centers has been shortened, but the by-produced HCl is increased. Meanwhile, a high LHSV can inhibit

water-gas shift reaction to $CO_2$ and $H_2$ ($CO + H_2O \leftrightarrow CO_2 + H_2$). However, $CH_4$ content in reforming gas increases with an increasing LHSV. As is well known, methanation reaction of CO or $CO_2$ is a reversible process ($CO + 3H_2 \leftrightarrow CH_4 + H_2O$ and $CO_2 + 4H_2 \leftrightarrow CH_4 + 2H_2O$). At a high LHSV, the formed $CH_4$ from methanation of CO or $CO_2$ easily runs off the catalyst surface. The reverse reaction such as steam reforming of $CH_4$ has difficulty being carried out.

**Table 3.** Effect of reforming parameters on catalytic property of KCl-modified NCML in steam reforming $CHCl_3$-EA *.

| Heating Rate during Catalyst Reduction ($°C·min^{-1}$) | LHSV ($h^{-1}$) | $H_2O/C$ Ratio | Conversion of Oil (%) | Capacity of Gas Production ($m^3 \, kg_{oil}^{-1}$) | Catalytic Life (min) | Gaseous Products Distribution (mol%) | | | |
|---|---|---|---|---|---|---|---|---|---|
| | | | | | | $H_2$ | CO | $CO_2$ | $CH_4$ |
| 2 | 3.30 | 2.13 | 96.24 | 3.95 | 460 | 66.38 | 12.89 | 14.94 | 5.79 |
| 5 | 3.30 | 2.13 | 97.37 | 4.26 | 450 | 67.34 | 13.32 | 14.88 | 4.46 |
| 8 | 3.30 | 2.13 | 97.83 | 4.35 | 408 | 69.83 | 12.85 | 14.78 | 2.55 |
| 10 | 3.30 | 2.13 | 98.16 | 4.27 | 412 | 69.56 | 12.53 | 14.89 | 3.02 |
| 5 | 6.66 | 2.13 | 83.64 | 3.04 | 190 | 66.19 | 16.77 | 11.14 | 5.90 |
| 5 | 4.40 | 2.13 | 90.20 | 4.03 | 280 | 66.84 | 15.62 | 12.04 | 5.51 |
| 5 | 3.77 | 2.13 | 94.96 | 4.11 | 430 | 68.10 | 15.06 | 12.52 | 4.31 |
| 5 | 3.30 | 4.21 | 97.96 | 4.55 | 1140 | 68.96 | 11.56 | 16.86 | 2.62 |
| 5 | 3.30 | 5.17 | 98.34 | 4.72 | 1620 | 69.07 | 11.36 | 16.97 | 2.60 |
| 5 | 3.30 | 8.11 | 98.64 | 5.48 | 3060 | 70.31 | 8.01 | 19.51 | 2.17 |

* Catalyst reduction at 800 °C, reforming temperature 750 °C, chlorine content of $CHCl_3$-EA 67,800 ppm, 80 $mL·min^{-1}$ $N_2$ as balance gas.

From the effect of $H_2O/C$ molar ratio on CSR of $CHCl_3$-EA over KCl/NCML catalyst, it can easily be discovered that increasing $H_2O/C$ ratio of the reforming system is beneficial to prolonging catalyst stability and an increase in $CHCl_3$-EA conversion, gas production capacity, and content of $H_2$ and $CO_2$ in reforming gaseous products. It is mainly because the introduction of more steam can accelerate the reforming of organic substances including $CHCl_3$, EA, and even $CH_4$ from the methanation, improve the gasification of the deposited carbon, and promote the water-gas shift reaction of CO to $CO_2$ and $H_2$. Moreover, the presence of more steam in the reforming system with a high $H_2O/C$ ratio will decrease the content of the by-produced HCl in the reaction system, which is helpful to protecting catalytic active sites. Meanwhile, at an $H_2O/C$ ratio of 8.11, the catalytic life can reach up to 3060 min, which is much longer than the 750 min catalytic life of the reported Ni-base catalyst [35,42].

Moreover, for KCl/NCML reduced at 800 °C with a heating rate of 5 $°C·min^{-1}$, it is found that the catalytic property of spent catalyst can be recovered to some extent after calcination at 750 °C in air. Under the reforming conditions of temperature 750 °C, LHSV 3.3 $h^{-1}$, chlorine content in organic waste liquid 67,800 ppm and $H_2O/C$ molar ratio 2.13, oil conversion of and gas production capacity reach 93.2% and 3.85 $m^3·kg_{oil}^{-1}$, respectively. The catalytic life can also reach up to about 400 min. However, the spent NCML without chlorinated salts modification exhibits a weak catalytic activity after the same regeneration. It shows that KCl modification protects the NCML reforming catalyst.

## 3. Materials and Methods

### 3.1. Catalysts Preparation

According to our previous work [41], Ni-Cu reforming catalysts supported $Al_2O_3$ modified by Mo and La as promoters were prepared through coprecipitation. Typically, nitrates of metals such as Ni, Cu, La, and Al were first dissolved in a desired amount of distilled water, the total concentration of all metal ions was kept to 1 $mol·L^{-1}$. The metal ion aqueous solution and $Na_2CO_3$-NaOH mixed alkaline solution containing $(NH_4)_6Mo_7O_2$ were simultaneously added dropwise into a reactor under stirring. Meanwhile, the pH value of reacted slurry during coprecipitation was kept at about 8.5 ± 0.2. The molar ratio of the introduced metal ions such as Ni, Cu, Mo, La, and Al was controlled to a value of 8:2:1:1.5:31. After finishing coprecipitation, the stirring was continued for another 2 h. The resulting slurry was aged overnight, filtered, and washed with distilled water till pH = 7. Finally, the filter cakes were dried at 105 °C, and calcined at 950 °C for 6 h in air atmosphere. The obtained samples were donated as NCML. The pre-introduction

of $Cl^-$ ions into catalysts was achieved by impregnation using $NH_4Cl$, NaCl, and KCl as chlorination agents, respectively. The introduced amount of chlorination agent was kept to 1 wt% Cl relative to catalyst solid. The impregnated samples were dried at 50 °C and calcined at 800 °C for 2 h. The obtained catalysts were donated as $NH_4Cl$/NCML, NaCl/NCML, and KCl/NCML, respectively.

### 3.2. Catalysts Characterization

$N_2$ adsorption–desorption of samples were carried out on an SSA-4200 micromeritics instrument (Beijing Builder Co., Beijing, China) at −196 °C. BET specific surface area of samples was calculated by the BET method according to the adsorption isotherms at the relative pressure (p/p$_0$) of 0.05–0.35, and pore volume was obtained by BJH method according to desorption isotherms.

Power X-ray diffraction (XRD) technique was used to analyze crystal composition of calcined and reduced samples, their XRD patterns were recorded on a DX-2700 powder diffractometer (Dandong Fangyuan Co., Dandong, China) using Cu K$\alpha$ radiation ($\lambda$ = 0.15406 nm) at 40 kV and 30 mA. The metal unit cell size of Ni metallic crystal for the catalyst was calculated by using the Scherrer formula.

Hydrogen temperature-programmed reduction ($H_2$-TPR), ammonia or $CO_2$ temperature-programmed desorption ($NH_3$-TPD or $CO_2$-TPD) were performed with a PCA-1200 adsorption instrument (Beijing Builder Co., Beijing, China) equipped with a thermal conductivity detector (TCD). Typically, 200 mg samples of $H_2$-TPR were pretreated at 300 °C for 30 min in 30 mL·min$^{-1}$ of He flow to remove adsorbed water and impurities. After cooling down to room temperature, samples were heated to 950 °C with a heating rate of 10 °C·min$^{-1}$ in 30 mL·min$^{-1}$ of 5 vol% $H_2$-$N_2$ mixture gas flow. The consumption of $H_2$ was detected by TCD. For $NH_3$-TPD and $CO_2$-TPD, 200 mg samples were first reduced by $H_2$ through heating at 10 °C·min$^{-1}$ to 800 °C and kept for 30 min. An amount totaling 30 mL·min$^{-1}$ of $NH_3$ or $CO_2$ was introduced for adsorption 30 min after sample temperature was cooled to 80 °C. After completing adsorption of $NH_3$ or $CO_2$, 30 mL·min$^{-1}$ of He was shifted to remove physically adsorbed $NH_3$ or $CO_2$ for 2 h. At last, the sample was heated in He at a rate of 10 °C·min$^{-1}$ from room temperature to 850 °C for $NH_3$ or $CO_2$ desorption. The desorbed $NH_3$ or $CO_2$ was also monitored by TCD.

Scanning electron micrographs (SEM) were obtained with ZEISS Gemini 300 scanning microscope operating at an accelerating voltage in a range of 2–3 kV, work distance of 4.9–5.1 mm.

### 3.3. Catalytic Test

The mixture of $CHCl_3$-EA with a molar ratio of 1:16.5 was used as the simulated organic waste liquid containing chlorocarbon. Steam reforming reaction was carried out in a fixed-bed quartz tube reactor (ø$_{in}$ 14 mm). 10 g dry catalysts (apparent volume about 15 mL) were loaded and reduced at 800 °C for 2 h in a 60 mL·min$^{-1}$ $H_2$ flow. Then, 80 mL·min$^{-1}$ of $N_2$ as balance carrier gas was switched to remove residual $H_2$ for 30 min, and the temperature was adjusted to 750 °C. $CHCl_3$-EA mixture liquid and distilled water were simultaneously injected into the catalyst bed by two HPLC pumps (Beijing Qingfang Co., Beijing, China) to start the reaction measurements. The volume of produced gaseous mixture was measured with a wet-type gas flowmeter, and the composition of gas products including $H_2$, CO, $CO_2$, and $CH_4$ was analyzed by an on-line gas chromatograph (GC-2000 Chongqing Sichuan Instrument, Chongqing, China) equipped with a TDX-01 packed column (carbon molecular sieves, 3 m × 3 mm) and TCD. The unreacted $CHCl_3$ and EA in the collected liquid mixture were analyzed by GC-3000 gas chromatograph equipped with a DB-5 capillary column and FID to calculate carbon conversion. The amount of deposited carbon on spent catalyst samples was measured according to weight loss after carbon combustion as the following formula: carbon deposit ($mg \cdot g_{Cat}^{-1} \cdot h^{-1}$) = ($W_{spent}$ − $W_{c,s}$)*1000/$W_{c,s}$/$T$, where $W_s$ represents the weight of spent catalyst (g), $W_{c,s}$ weight of spent catalyst after calcination at 750 °C (g), $T$ time on stream for reforming.

## 4. Conclusions

Chloride salts are introduced into Ni-Cu-Mo-La catalysts in order to prevent adsorption and reaction of produced HCl during steam reforming of chloroform-ethyl acetate mixture. The resistance to Cl-poisoning and a good stability of reforming catalyst are hoped to be improved greatly. It is proved that the pre-introduced Cl$^-$ ions exhibit a synergic effect on the improvement of catalytic stability in the steam reforming that occurs in the catalytic active sites of the Ni-Cu metallic phase. The destruction of catalyst structure caused by the produced HCl and carbon deposits are two important factors affecting the catalytic life of catalysts in the steam reforming of CHCl$_3$-EA. KCl-modification increases the catalytic life of NiCuMoLaAl reforming catalyst from 208 min to 408 min, which can be further prolonged up to 3060 min when decreasing chlorine content in organic reactant and increasing the introduced amount of steam. Meanwhile, the catalytic performance of the deactivated KCl/NiCuMoLaAl can be recovered to some extent after regeneration through calcination at 750 °C in air. All evidence shows that KCl/NiCuMoLaAl is a potential catalyst for resource utilization of chlorocarbon-containing organic waste liquid to hydrogen or syngas.

**Author Contributions:** Conceptualization, Q.W. and C.X.; writing—original draft preparation, Q.W.; writing—review and editing, C.X.; supervision, C.X., discussion and analysis, Y.Z., J.L. (Jie Liu), Z.D., and J.L. (Jianying Liu); All authors have read and agreed to the published version of the manuscript.

**Funding:** This research was funded by International Corporation Projects, the Science and Technology Department of Sichuan Province (2019YFH0133), and the Education Department of Sichuan Province (14CZ0020).

**Data Availability Statement:** Data is contained within the article.

**Acknowledgments:** We would like to thank Mengdie Li from Shiyanjia Lab for her help with the SEM experiments.

**Conflicts of Interest:** The authors declare no conflict of interest.

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
