# Peer review of "Steam Reforming of Chloroform-Ethyl Acetate Mixture to Syngas over Ni-Cu Based Catalysts"

_catalysts, doi:10.3390/catal11070826_

Round 1
Reviewer 1 Report
The authors studied the steam reforming reaction of chloroform-ethyl acetate mixture via a series of Ni-Cu catalysts modified with different salts all containing chlorine. They showed that the modification with KCl gives the best catalytic activity, particularly in terms of stability.
The topic of the work is timely as the development of poison-resistance catalysts for the sustainable hydrogen production is imperative for the ecological transition. The work may be published after addressing the following issues:
- Steam reforming of Cl-containing streams is quite old and has been practically abandoned for the last 15-20 years or so. The main reason is that Cl destroys the catalysts and even the addition of salts can prevent that--for instance, Cl forms liquid salts with Ca at the steam reforming reaction conditions which totally kills the catalysts. In light of the fact that the authors here report only a very minimal gain in catalysts lifetime with the KCl modification, basically adding very little knowledge on the topic, could they better argument on why this paper should be published?
- I have many concerns about Section 2.1:
- Rows 83-84: How can the authors state that the catalytic activity is high? Is there a way to compare their results with some others?
- Rows 90-91: How can the authors be so sure that the introduction of ions weaken the catalytic activity? Could it simply be due to a lower metallic active phase exposure? Please, explain.
- Rows 94-101: Before drawing conclusions on both water-gas shift and methanation reactions, it would be better to run them with the very same catalysts. If that is not possible, please remove this part as the statements there reported cannot be supported by any catalytical data.
- Rows 112-133: Actually I do not see the "sharp stop". Please add the full data in Figure 1.
- Rows 120-124: In general, carbon deposition is a gradual process which does not lead to the sudden catalyst failure. Considering also the very short time frame of the reactions here reported, how can the authors be sure that the reason behind the (not shown) catalysts failure is carbon deposition? Please clarify.
- In Table 1 coke deposition rates are reported but it is not clear to me how were calculated. Please clarify.
- Section 2.2: Row 147: If the phase quantification through XRD is not possible, but would be desirable, please refrain from referring to "amounts" when discussing XRD data.
- In Section 2.3:
- Although the conclusions drawn by the authors about the catalysts reducibility and, in turn, the catalysts phase composition, might be sound, it would be best to calculate the hydrogen actually consumed during the reduction events. This will surely provide much more solid insights which species in reducing. I say so because from the H2-TPR analyses of Figure 3, one might infer that the Ni and Cu exist as totally separated species in the not modified catalysts since it is well known that Cu reduces at low temperatures while Ni at very high temperatures. The addition of Cl salts my then lead to the formation of Ni-Cu alloys, as also evidence by the XRD data, which might be reduced at an intermediate temperature, such as indeed 340 °C.
- The decrease in reduction temperature observed by the authors could be due to the addition of alkali metals, such as Na and K, which are very well known reduction promoters. Could this be a reason for the (slightly) improved catalysts lifetime?
- Rows 208-209: How did the authors come with a core-shell structure? No previous and neither later data corroborate this, so please clarify. HRTEM images would be helpful in this sense.
- In Section 2.4: Rows 226.231: How are the CO2-TPD data correlated to the work? I mean, alkali metals, apart from favoring the reduction of metals, are also well known to introduce basic sites. They are generally useful to prevent the coke deposition, but here it seems no effect is exerted. Please calrify.
- Section 2.5
- Rows 242-243: How could the authors see uniform nanoparticles through low magnification SEM images? Please remove any comment on the nanoparticles morphology from this section.
- Rows 247-250: Maybe it is just me, but I thought that the main cause for the catalysts failure was carbon deposition, whereas now it is, at least for one of them, the catalysts destruction. Please clarify.
- In Section 2.6:
- Row 265: I could not find any Table 3.
- Often the authors talk about "Cu cores" and Ni--Ni interphases which in my opinion were not demonstrated to exist. Please remove any references to those.
- Rows 277-279: How can the authors be so sure that the increased CH4 selectivity is solely due to methanation at higher space velocity? It might also be due to incomplete reforming of the reactants. Again, if no such specific reactions are carried out, please do not draw hasty conclusions.
- Rows 284-286: The increase of the steam-to-carbon ration is known to facilitate the steam reforming reactions and improve the coke gasification. How can the authors be so sure that the high steam content prevents the HCl to interact with the catalytic sites? Could it be that the HCl forms on the very reforming catalytic sites and so the more steam simply helps gasifying the coke? Please clarify.
Reviewer 2 Report
In this paper, the authors prepared NiCuMoLaAl mixed oxide catalysts which were tested in steam reforming of chloroform-ethyl acetate (CHCl3-EA) mixture to syngas.
The paper, which is properly divided in sections and sub-sections, needs some corrections before its publication in the journal.
- The authors should extend the literature survey by adding more recent papers regarding the Ni-based catalysts for steam reforming. Some examples are: 10.1016/j.cep.2017.08.010, 10.3390/en10071030 and 10.3390/catal9080688;
- The authors should specify which are the reactions involved in the investigated process;
- The authors should add the thermodynamic equilibrium data in table 1;
- The authors should clarify in which a way they calculated the LSHV (maybe LHSV?) of table 1;
- The data reported in figure 1 evidenced that the capacity of gas production for the NaCl/NCML sample increased after about 130 minutes, differently from the other catalysts. Can the author better explain this behavior?
- Did the authors perform thermal analysis (such as Thermogravimetric analysis) in order to further investigate the nature of the coke deposited on the samples? This aspect could be useful for example in order to assess the regeneration procedure of the catalysts;
- Regarding this last aspect, did the authors perform any consecutive cycles of reaction and regeneration of the catalysts, in order to further investigate their behavior?
- The authors should introduce the acronyms when they use them for the first time in the text (for example LSHV);
- The authors should express the equilibrium reaction in such a way, for example the WGS reaction at line 276;
- The authors should compare the performance of their catalysts with the ones of other catalysts in literature;
- The authors should better summarize the main results of their research in the conclusions section;
Round 2
Reviewer 1 Report
The authors answered only to some of the issues raised during the first round of reviewing. Taking the bulleting from the authors' answer report:
- Point 3: The metallic phase exposure should be first calculated and only then can be involved as reason for the lower catalytic activity. If that is not possible, the phrase "It shows that the introduction of ions reduces exposure metallic active phase of reforming catalysts" is pure postulation and as such must be eliminated.
- Point 4: In theory I agree with the authors about their considerations on the WGS and methanation reactions. But since no solid evidence on the "weight" of such reactions on the overall catalytic activity is provided, please specify that these are only speculations.
- Point 5: I find highly unlikely that the catalytic activity can plummet to zero with a perfectly vertical line, as those in the new Figure 1. As I already said in the first round, catalyst deactivation by coking is a progressive drop in the activity, not a line with infinite slope. Please explain.
- Point 6: A follow up of the previous point: If the catalytic activity really zeroed in a such short timeframe, I hope the authors took all the necessary safety precautions. A reactor at high temperature suddenly filling up with a water mixture containing chlorinated salts-and possibly HCl-is a major concern. Can the authors comment on the safety of such a reaction?
- Point 9: I totally do not agree with the authors' on this point. Since no H2 consumption has been reported, and since no other characterizations were performed that might have shed some light on the metallic phases, such as EXAFS or XANES, the comments on the H2-TPR analyses are purely speculative. Either the authors clearly state that or please remove them entirely from the manuscript because they can mislead the readers.
- Point 10: The authors argue that the increased catalytic activity is also due to "the presence of Cl ions in catalysts [that] prevent the
adsorption and interaction HCl with active phase, form less acidic sites". Apart again from the fact that no evidence has been provided to support such claim, how the authors correlate it to the apparent increase in basic sites in the Cl-modified catalysts with which HCl may easily interact? If no HCl adsorption analyses can be run, please remove any statement on HCl adsorption. - Point 12: The authors state that "The easy reduction of metals mainly comes from Cl ions". Can the author provide evidence of such claim or reference to the literature?
Reviewer 2 Report
The authors in this revised version addressed the comments of the reviewer, but some minor issues are still present. In particular, the authors should check the reference list: the references 22, 23 and 27 are wrongly written, since the authors reported the name instead of the surname of the authors. For example Vincenzo P. is Palma V., or Gianluca L. is Landi G., and so on
Round 3
Reviewer 1 Report
I thank the authors for answering to each and every issue with precision. It is unfortunate that most of the conclusions of this work are based on speculation since only marginal experimental evidence, and none theoretical, are provided.
I still have some concerns about the catalytic tests. Since the very authors agree that catalyst failure by coking is a progressive process and does not look like a sudden drop, I wonder what is happening to these reactions. Just to shed any further doubts, can the authors report the actual outlet gas flow rates per each catalytic test? If the reactor and/or the catalyst bed is progressively coking up or being "destroyed" by the allegedly HCl production, a slow decrease of the total flow rate should be observed. If that won't be the case, it would imply that until the very last catalytic site is covered and/or "destroyed", the catalytic activity remains unaltered. This in turn would mean that all the other catalytic sites are useless and, as such, the soundness of this work should be questioned.
